# Therapeutic Potential of Dental Pulp Stem Cells According to Different Transplant Types

**DOI:** 10.3390/molecules26247423

**Published:** 2021-12-07

**Authors:** Tomasz Staniowski, Anna Zawadzka-Knefel, Katarzyna Skośkiewicz-Malinowska

**Affiliations:** Department of Conservative Dentistry with Endodontics, Wroclaw Medical University, 50-425 Wrocław, Poland; tomasz.staniowski@umw.edu.pl (T.S.); katarzyna.skoskiewicz-malinowska@umw.edu.pl (K.S.-M.)

**Keywords:** endodontics, dental pulp stem cells, scaffold, growth factor

## Abstract

Stem cells are unspecialised cells capable of perpetual self-renewal, proliferation and differentiation into more specialised daughter cells. They are present in many tissues and organs, including the stomatognathic system. Recently, the great interest of scientists in obtaining stem cells from human teeth is due to their easy availability and a non-invasive procedure of collecting the material. Three key components are required for tissue regeneration: stem cells, appropriate scaffold material and growth factors. Depending on the source of the new tissue or organ, there are several types of transplants. In this review, the following division into four transplant types is applied due to genetic differences between the donor and the recipient: xenotransplantation, allotransplantation, autotransplantation and isotransplantation (however, due to the lack of research, type was not included). In vivo studies have shown that Dental Pulp Stem Cells (DPSCs)can form a dentin-pulp complex, nerves, adipose, bone, cartilage, skin, blood vessels and myocardium, which gives hope for their use in various biomedical areas, such as immunotherapy and regenerative therapy. This review presents the current in vivo research and advances to provide new biological insights and therapeutic possibilities of using DPSCs.

## 1. Introduction

Stem cells are unspecialised cells capable of perpetual self-renewal, proliferation and differentiation into more specialised daughter cells [1]. They are present in many tissues and organs, including the stomatognathic system. There are three basic types of these cells: embryonic stem cells (ESCs), induced pluripotent stem cells (iPSCs) and adult stem cells (ASCs) [2]. The embryonic stem cells are isolated from the blastocyst during embryonic development; they are pluripotent and can transform into many types of cells derived from all three germ layers: ectoderm, mesoderm and endoderm. The collection method and harvesting cause an ethical issue and difficulties in obtaining these cells, and iPSCs are generated artificially from somatic cells. Due to the ethical issues, teratoma formation and immune responses of embryonic and induced pluripotent stem cells, most of the current research is focused on adult-derived stem cells. In our review, we present the use of Dental Pulp Stem Cells DPSCs, taking into account each type of transplant. In vivo experiments can be divided into xenotransplantation, i.e., the transplant of cells or tissue from one species to another; allotransplantation, i.e., the transplant of an organ, tissue or cells between two genetically non-identical members of the same species; and autotransplantation, i.e., the transplant of tissue to the same person.

Mesenchymal stem cells (MSCs) can be isolated from human and animal sources, including bone marrow, adipose tissue, epidermis, muscle, liver, amniotic fluid, placenta, umbilical cord blood, menstrual blood and dental pulp [3]. MSCs demonstrate beneficial effects in regenerative and transplantation medicine. They show therapeutic potential, exhibiting multipotent properties and the capacity to differentiate across more than one cell line under normal circumstances. Depending on the type, they have different abilities to differentiate (Figure 1).

MSCs populate stem cell niches—specialised microenvironments that provides structural and functional cues that are both biochemical and biophysical—and stem cells integrate this complex array of signals with intrinsic regulatory networks. Stem cells remain inactive in the niches until the stimulus is triggered—indicating a damaged organ or physiological demand—leading to differentiation into specialised cells that build into the damaged tissue, causing its regeneration and renewal [4].

After isolation, expansion and in vivo application, MSCs can penetrate damaged tissues and modulate the inflammatory response through synergistic regulation—reduction of pro-inflammatory cytokines and enhancement of both survival factors and anti-inflammatory factors [1]. MSCs interact with most cell types of the innate and acquired immune system, including B cells, T cells, dendritic cells (DCs), natural killer (NK) cells, neutrophils and macrophages, moderating their response to pathogens [5]. They are used in immune/non-immune/gene therapies and for tissue repair injury, trauma, ischaemia, burns, radiation and degenerative diseases [6]. They have been studied extensively for lung pathology. Due to immunomodulatory and anti-inflammatory properties and regenerative stem cells, research on their use in the treatment of COVID-19 is ongoing [7]. The research group reported that the fibroblast growth factor (FGF) isolated from MSCs exhibited viral replication inhibition properties [5].

The International Society of Cellular Therapy has outlined the main criteria for determining multipotent human mesenchymal stromal cells. These are: MSCs must adhere to plastic when maintained in standard culture conditions using tissue culture flasks; ≥95% of the MSC population must express CD105, CD73 and CD90 as measured by flow cytometry and lack expression (≤52% positive) of CD45, CD34, CD14 or CD11b, CD79a or CD19 and HLA class II; and the cells must also differentiate in vitro into osteoblasts, adipocytes and chondroblasts under the influence of a specific stimulus [8]. In vitro differentiation of MSCs depends mainly on the presence of growth factors and cytokines in the growth medium.

## 2. Characterisation of Dental Pulp Stem Cells

DPSCs are multipotent and can differentiate into three distinct cell lines, such as odontoblasts, osteoblasts, endothelial cells and nerve cells [9]. The most common sources for human DPSCs harvested are wisdom teeth or first premolars. Research shows that dental pulp stem cells have a higher therapeutic value in the treatment of spinal cord injuries compared to the treatment of human bone marrow-derived MSCs (BMSCs) [10]. 

The primary function of the pulp is the formation of primary dentin during tooth development; then secondary dentin, which develops with age; and tertiary dentin, which is a response to pathological factors. The tooth pulp is surrounded by mineralised structures. The only connection with the surrounding tissues is in the anatomical opening of the root apex, where blood vessels and nerves run. It is the only way to nourish the pulp and periodontal complex and remove waste products. Morphologically, in the pulp, we can distinguish the outermost layer of odontoblasts, cell-free zone (zone of Weil), cell-rich zone and the pulp core, which is the centre of the pulp [11]. The developing pulpitis, associated with exudate, has a limited ability to regenerate and eliminate microorganisms and toxic substances. The outer layer of the pulp cells are odontoblasts specialised in producing the extracellular matrix (ECM), its mineralisation and appropriate structure formation. In addition, odontoblasts secrete defence substances: interleukin-13 (IL-13), interleukin-6 (IL-6), tumor necrosis factor (TNF) and chemokines in response to irritants [12]. 

In the case of pulp irritation caused by slowly progressive tooth decay, tooth wear, short-range non-carious cavities (erosion, abrasion, abfraction) or slight fracture of the tooth tissues, odontoblasts are stimulated to produce dentin under the irritating factor. The study shows the importance of a heparin-binding EGF-like factor (HB-EGF), transforming growth factor beta-1 (TGF-β1), transforming growth factor beta-3 (TGF-β3), insulin-like growth factor-1 (IGF-1), insulin-like growth factor-2 (IGF-2) and angiopoetin-related growth factor (AGF) to in vitro dentinogenesis [13,14].

If the tooth tissue damage is greater or progresses more rapidly, the odontoblast layer is destroyed. Eventually, the differentiated odontoblast cells lose the ability to multiply and regenerate the new layer. Primary odontoblasts are replaced by odontoblast-like cells that distinguish themselves from the proliferation of progenitor cells (DPSCs). Their reservoir is located around the perivascular niches in the pulp [4]. Changes in the dental pulp during ageing (decreased vascular supply, modification of the niches and obliteration of pulp chamber) affects function of DPSCs. These symptoms can affect the regenerative capacity of stem cells [15]. Further studies of changes in dental pulp stem cells with age may play an important role in their use in the elderly. The undifferentiated stem cells of the dental pulp can differentiate into odontoblasts, fibroblasts and other cells necessary for pulp functioning and regeneration. This may occur under the influence of various growth factors and odontotropic preparations (calcium hydroxide, HA and MTA) [16]. The ability to stimulate the stem cells of the tooth pulp into odontoblasts has the greatest importance for its regeneration [17,18]. DPSC morphology represents their photosynthetic and metabolic activity. These cells have a spindle shape similar to fibroblasts. However, fibroblasts contain larger and more numerous nucleoli, a better-developed rough endoplasmic reticulum as well as longer and more branched filopodia. DPSCs are distinguished by centrally arranged spherical or irregularly shaped large pale nuclei (often eccentrically located) with a large amount of euchromatin, Golgi apparatus and abundant cisterns of rough endoplasmic reticulum and numerous coated matrix vesicles. Due to many thin plasmalemma processes, microvilli and filopodia adhere to the substrate [19]. Stem cells isolated from the pulp tissue of third-molar-impacted teeth were described in 2000 by Gronthos et al. [20]. Comparing the properties of DPSCs and MSCs, they showed the promineralisation nature of DPSCs due to the suppression of specific markers that are characteristic for bone tissue, including collagen type I, calcium phosphatase, osteopontin, osteonectin and osteocalcin. A distinguishing feature of DPSCs is the ability to regenerate the pulp-dentin complex, as evidenced by the ability to produce dentin sialoprotein.

The oral cavity is a rich source of stem cells; so far, iPSC cells have been isolated from dental pulp stem cells (DPSCs), stem cells from human exfoliated deciduous teeth (SHEDs), periodontal ligament stem cells (PDLSCs), stem cells from apical papilla (SCAPs) and dental follicle progenitor cells (DFPCs) [21]. All of them (except SHEDs) come from permanent teeth and belong to the group of mesenchymal stem cells. Despite their different origins, all of them demonstrate a self-renewal ability and multilineage differentiation potential. The regeneration/revascularisation of pulpal tissues uses DSCs in partnership with growth factors, scaffolds and vascular supply [22].

### 2.1. Surface Markers

Distinguishing stem cells is possible due to the presence of surface markers, specific proteins that are receptors with the capacity to bind or adhere to another molecule selectively. The most common surface marker molecules used to describe MSCs are CD105, CD271, CD44, CD73, CD90 and STRO-1, while the negative ones are CD34, CD45 and HLA-DR. However, it is difficult to isolate those that are specific only to DPSCs [23].

Analysis of the researchers’ reports showed that many surface markers in the DPSC population are determined, but not every DPSC population simultaneously expresses all surface markers because the properties of different populations may differ from one another. DPSC surface markers can be divided into those most often determinable for MSC, i.e., STRO-1 [24], CD29, CD44, CD73, CD90, CD105, CD146, CD166 and CD271 [25,26,27,28], as well as markers CD34, CD117 [29,30], OCT-3/4 and NANOG [31,32].

STRO-1 is a well-known marker of MSCs, and its expression in the pulp of the tooth is demonstrated mainly by pericytes [33]. This marker can be a useful tool to identify hard tissue formation, which is due to the tendency of STRO-1 + DPSCs to differentiate into odontoblasts and osteoblast-like cells [34,35,36]. The presence of the STRO-1 receptor may indicate the origin of DPSCs from perivascular niches due to the location of this receptor not only in the blood vessels in the tooth pulp but also in the large blood vessels, but not capillaries. There is also a supposition that DPSC cells may exist in the perineural spaces, as evidenced by the presence of a connection between the periceum of the tooth pulp and peripheral nerve bundles and the STRO-1 receptor [33]. 

Other well-known MCS markers, apart from STRO-1, are CD29 and CD44, both also identified as DPSCs markers [26,32,37]. CD29 from the β1 integrin family and CD44 are integral cell membrane glycoproteins that play a role in activating the adhesion of lymphocytes to the matrix and lymph nodes [38]. In addition, the expression of CD73 and CD90 has also been observed in DPDCs. CD90 (Thy-1) is a glycoprotein involved in cell–cell and cell–matrix interactions, while CD73 mediates the binding of lymphocytes to endothelial cells [25,38]. Another marker localised in DPSC cells is CD105, associated with the human vascular endothelium. This marker is a component of the transforming growth factor-beta receptor (TGFbeta) complex and links TGFbeta-1. DPSC cells that express CD105 show high proliferative and migratory capacity [26,39].

The marker CD146 is found in human endothelial cells and is shown by many authors as a marker expressed by DPSCs, similar to CD166 observed in human DPSCs.

The CD271 marker is one of two types of receptors for neurotrophins; it belongs to the group of protein growth factors that stimulate nerve cells (to survive) and is also a marker of the MAC cells of the tooth pulp. There are also reports that this marker was associated with inhibiting MSC differentiation, including DPSCs, in osteogenic, adipogenic, chondrogenic and myogenic cell lines [25,26,28].

DPSCs can also express bone, dentin and cementum (bone morphogenic protein (BMP), alkaline phosphatase, osteonectin, osteopontin and bone sialoprotein) and fibroblasts (collagen type I and III) [22]. In addition, studies have shown that DPSCs can express toll-like receptors to recognise bacteria, TLR4, TLR2 and the vascular endothelial growth factor in response to lipopolysaccharide—a product of gram-negative bacteria, which plays a vital role in response to invasions of pro-cariogenic bacteria [40].

### 2.2. Isolation of DPSCs

Obtaining stem cells from the tooth pulp is much easier than from other tissues, especially bone marrow; however, it produces several problems. The first is the time between the tooth extraction and the gain of cells; the second is the storage of the tooth; the third is the tooth surface disinfection protocol; and lastly, the method of pulp removal from the chamber. Human DPSCs are harvested most often from extracted wisdom teeth [20], deciduous teeth [41], apical papilla [42], supernumerary teeth [43] and teeth with broken tooth crowns [44]. According to reports, the inflamed pulp can also be a source of DPSCs, and their properties are similar to those removed from the living, healthy pulp [25,45].

The time between the tooth removal, the possibility of gaining cells and the conditions for their storage are important for obtaining DPSCs. In the case of storing teeth in PBS (phosphate-buffered saline) or HTS (HypoThermalsol) at 4 °C, it is possible to receive stem cells up to 120 h after extraction [46]. Teeth can be transported in Minimum Essential Medium Eagle—Alpha Modification (α-MEM) [47]. In the studies of Alkhalil et al., out of 19 extracted teeth, only three pulp stem cell lines were isolated. Extracted teeth were placed in sterile tubes for 6–18 h at 0–4 °C overnight. After that, the dental pulp was extirpated, digested, centrifuged and incubated in DMEM/Low Glc with FBS at 37 °C and 5% CO_2_. During the isolation of the pulp stem cells, the time between pulp extraction and extirpation and the start of cultivation is critical. The authors suggest that taking too long (10 h) between these two procedures causes the death of DPSCs and other cells [48].

Extracted teeth should be washed using the disinfection protocol. For this procedure, different liquids can be used, including sterile saline, polyvinylpyrrolidone-iodine, 0.1% sodium thiosulfate, Listerine^®^, followed by several final washes in sterile PBS [49], solutions containing antibiotics, chlorhexidine gel [50], chlorhexidine gluconate and Listerine^®^ mouthwashes [51] and 70% ethanol [47].

Dental pulp after teeth extraction can be extirpated after fracturing the dental crown into parts using pliers (bone forceps) [37] or cutting using handpiece and diamond burs and producing an incision at the boundary of the enamel–cement junction [47,52].

### 2.3. DPSC Culture

The two main methods were used to culture dental pulp stem cells. The enzyme digestion method is the first one, and the explant outgrowth is the second. Both of them were described and compared by Huang et al. (Figure 2). After extraction of third molars from healthy patients, pulp cell cultures were established via two approaches: (1) enzyme digestion was carried out. First, pulp tissues were digested using a solution of collagenase type I and dispase. Then, digested cell suspensions passed through a cell strainer. Single-cell suspensions were seeded under 5% CO_2_ at 37 °C in culture flasks containing α-MEM supplemented with FBS and a mixture of l-glutamine, l-ascorbic acid-2-phosphate and antibiotics. Pulp tissue small parts (2 × 2 × 1 mm) were placed in plates with DMEM supplemented with FBS and antibiotics. The outgrown cells were transferred to culture flasks and grown to confluence. The authors noticed that the cell isolation method plays an important role in the obtained cell population distribution [53]. The same method was used to isolate DPCs and compare gene expression DPCs with MSCs (derived from bone marrow, synovium and adipose tissue) and fibroblasts (FBs), osteoblasts, adipocytes and chondrocytes [54]. Enzymatic digested DPSCs compared with outgrowth ones indicates higher mineralisation capacity [47]. 

Raoof and al. compared the effectiveness of three methods of obtaining stem cells. After extraction, teeth were stored in ice-immersed phosphate buffer saline. Then they were cut with a high-speed handpiece using a diamond fissure bur with water supply following the cementoenamel junction. Two of them were similar to those described above. The three groups of teeth were done: (1) pulp pieces were digested by collagenase or dispase enzyme, and released cells were cultured; (2) pulp pieces were undigested and cultured; (3) pulp pieces were digested and fixed. Tissues were digested in collagenase or dispase solution. Cell suspensions were incubated at 37 °C and 5% CO_2_ in culture dishes containing minimum essential medium (α-MEM) with 20% FBS in a mixture of antibiotics. The cell viability was determined by trypan blue. The authors demonstrated that the third method was better (more than 60%), and the second method was the worst in stem cell isolation [55]. DPSCs can be cultured in Dulbecco’s Modified Eagle Medium (DMEM) or α-MEM [52].

Deciduous teeth are a vital source of stem cells. After extraction, dental pulp tissues are digested in collagenase type I for 1 h. Next, the cell suspension is filtered, and single cells are cultured in DMEM with FCS at 37 °C and 5% CO_2_. Then, after reaching optimal confluency, cells are collected for further passages [56].

### 2.4. Methods of Stem Cells Characterisation 

Biochemical and molecular characterisation is necessary to confirm the obtaining of stem cells. The most common method used in identifying stem cells is colony-forming assay, an in vitro assay based on the ability of a single cell to form a colony of morphologically and functionally identical cells building a given tissue or specialised in performing a specific function. The colony must consist of at least 50 cells [57].

-Inverted phase-contrast microscopy images of the morphologies and behaviour of acquired cells-Quantitative real-time polymerase chain reaction (qRT-PCR) provides a flexible, sensitive and scalable method for analysing the gene expression profile of any cell population. This method is used to duplicate the DNA fragments. With the help of the polymerase chain reaction (PCR) technique, we can obtain billions of copies of a specific DNA sequence in a short time. Then, we can determine its length and exact structure. The quantitative real-time polymerase chain reaction (qRT-PCR) method enables the measurement of the number of new copies of the reaction product at any point in time (in real time). It is a very sensitive technique that enables the detection of even one copy of the studied gene. It differs from traditional PCR in that it analyses the increase of the PCR reaction products after each reaction cycle, not only after the last one. Due to its specificity, the qPCR method has become a valuable diagnostic tool. By this method, we can identify numerous human MSC-characteristic genes [58].-Flow cytometry based on fluorochrome-conjugated antibodies against specific markers on the cell surface. Flow cytometry enables the analysis of several parameters for each cell, cell fragments, or other artifacts as the colour intensity, size and fluorescence of the tested cells. The results of the diagnosis are histograms, which allow us to visualise the populations of cells with specific characteristics in the tested sample. Cytometers equipped with a device for sorting cells in an electric field make it possible to evaluate some functional parameters of cells and isolation-appropriate cells [59]. -Immunofluorescence is used to confirm the presence or absence of protein expression in stem cells. Complexes resulting from combining antibodies with antigens are detected using fluorescent dyes.-Protein extraction and immunoblotting: a technique that uses specific antibodies to identify separated proteins based on size by gel electrophoresis.

### 2.5. Cryopreservation

During cryopreservation, important roles are played by the tissue type, freezing time and freezing parameters to obtain stem cells. With relatively little effort, optimal results were achieved with the pulp frozen directly after its removal from the tooth chamber. Then, only after thawing was digested, DPSCs were cultured. Freezing cultured DPSCs requires time-consuming initial work, but the results were not satisfactory for intact teeth. Cryopreserved agents (CPA) such as ethylene glycol (EG), propylene glycol and dimethyl sulfoxide (Me2SO) can be used to freeze DPSCs at different concentrations. Me2SO at a concentration between 1–1.5 M has the best properties. The temperature −85 °C or −196 °C and the DPSCs concentration at least up to 2 × 106 cells/mL caused no difference in cell culture after defrosting [49].

Dental pulp stem cells can be cryopreserved without losing their properties after post-thaw for one week [60], three months [61], one year [62] or two years [51]. The viability of dental pulp cells after two years of cryopreservation was 85% and kept appropriate differentiation capacity [51].

## 3. Characterisation of Scaffolds and Growth Factors

Selection of the appropriate scaffold, growth factors, chemicals or compounds with differentiating properties is essential to induce proliferation and differentiation of DPSCs in a particular direction in vivo. The application of physical and chemical signals, structural support and ensuring microenvironmental homeostasis enables the interaction and integration of stem cells after transplantation. The advancement of tissue engineering leads to the improvement of the possibilities of future development of cell sources through individual adaptation of cell supports, immune modulation, vascularisation and predictive abilities of computer and mathematical modelling for more complex materials [63]. Attempts are made to reproduce niches more and more accurately with their complex and dynamically changing interactions that regulate differentiation, survival and self-renewal of stem cells.

Various strategies were applied to stimulate stem cells to produce specific tissues and organs. In tissue engineering related to stem cells, the triad aimed at recreating the environment in vivo is widely used, mimicking the composition of the extracellular matrix (ECM) based on the use of a three-dimensional scaffold or biomaterials with implanted stem cells and stimulating factors—appropriate morphogenic, regulatory signals to differentiate into the required tissue type [64].

The factors stimulating the formation of new tissues include, among others, L-ascorbic acid, dexamethasone and β-glycerol phosphate supplementation when differentiated into osteoblast and odontoblasts; dexamethasone, 3-isobutyl-1-methylxanthine, insulin and indomethacin when differentiated into adipogenic cell lineages; dexamethasone, L-ascorbic acid, ITS, L-proline, sodium pyruvate and TGF-β3 when differentiated into chondroblasts; and BDNF, bFGF, B27 supplement, EGF, NT-3, NGF, forskolin, cyclic adenosine monophosphate, 3-isobutyl-1-methylxanthine, sonic hedgehog and retinoic acid when differentiated into neuron-like cells, dopaminergic neurons, oligodendrocytes and Schwann cells [64].

Although traditional 2D monolayer culture on a flat surface often provides an appropriate representation of cell behaviour, there are sometimes difficulties in recreating a natural microenvironment similar to that found in a living organism. The differences may alter gene expression patterns, cell shape or cell–cell interactions [65]. Studies have shown that although 2D culture allowed the production of cells, such as osteocytes, chondrocytes, adipocytes, cardiomyocytes, hepatocytes and neural cells, etc., those guided by specific growth factors during cell growth in vitro are not possible to reproduce the physiology of in vivo conditions accurately. Moreover, it was shown that multiplication of MSCs in monolayers changes their phenotype—broad and flattened morphology—which in turn changes cell fate and differentiation potential. Due to the difficulties of 2D culture, attempts were made to obtain a three-dimensional microenvironment using various biomaterials under static and dynamic conditions [64,66]. 

Creating an appropriate scaffold plays a key role in controlling the adhesion, differentiation and proliferation of DPSCs because they contain biological signals that allow them to act as a bioactive platform [64,67]. Furthermore, this three-dimensional scaffold space with numerous micropores is a place for transporting nutrients, diffusion of waste products and creating an environment with specific physico-chemical parameters [68], enabling proper functioning and providing living conditions similar to those in environmental niches of the living organism.

A three-dimensional tissue template should be made from a material that ensures proper cell attachment and adherence, migration, proliferation and differentiation, without a cytotoxic effect on tissues during the degradation process and without stimulating the host’s immune response [67].

Different types of natural and synthetic biomaterials are applied as a specific scaffold for each tissue that needs to be restored. 

All of them share certain features: biodegradability and biocompatibility and their use allows for more precise spatio-temporal targeting and differentiation of cells into specific cell lines [66].

Scientists have shown that interactions between biomaterials and stem cells can influence their differentiation even in the absence of delivered biomolecules [69]. It is possible to apply molecular, genetic, chemical and topographic cues to guide the fate of stem cells in vivo [70].

Numerous materials and their modifications are used to create a stable microenvironment for DPCSs. Among the biomaterials used in tissue engineering, the most common are synthetic polymers, including polylactic acid (PLA), polyethylene glycol (PEG), polylysine (PLL), polycaprolactone (PCL), polyglycolic acid (PGA), polycaprolactone (PCL) and poly-dl-lactic acid-co-glycolic acid (PGLA), natural hyaluronic acid (HA), chitosan, collagen type I (Col1), alginate, agarose, fibronectin and hydrogels produced by either synthetic polymers or natural polymers or hybrid polymers [71]. They also include ceramics, such as beta-tricalcium phosphate (β-TCP), hydroxyapatite (HA) and calcium silicate bioactive glass, as well as composites combining two or more materials, including co-polymers, polymer-polymer blends or polymer-ceramic composites [64].

There are numerous requirements for a scaffold in addition to biocompatibility, biodegradability and mechanical properties. Moreover, the material for the scaffolding should have an appropriate microstructure to generate useful signals and promote the formation of an appropriate tissue. For example, for the reconstruction of bone tissue, materials with a porous structure are most often selected—calcium phosphate glass bioactive polyesters or natural particles of collagen gels [72].

The use of flexible hydrogels coated with collagen type I stimulates the increase of myogenic markers, while the rigid gels imitating bone have an osteogenic effect. In the research of Griffin [70], it was shown that nanotopography alone mapping the three-dimensional features of the extracellular matrix (ECM) can induce stem cell stimulation. DPSCs respond to the alteration of nanotopography by altering gene expression, adhesion, migration, differentiation and proliferation [73].

The use of scaffolds is sometimes a challenge, most often due to the properties of the building substances. For example, ceramics and bioactive glass materials have certain limitations due to weakness and brittleness [74]. In the case of synthetic materials, there is a risk of rejection by the body or creating an unfavourable microenvironment conducive to tissue degradation [72,75]. In contrast, in the case of natural scaffolds, limitations are related to controlling the degradation rate and poor mechanical properties [76]. 

The challenge is also to limit in vitro cell differentiation by many factors. The analysis of biochemical features often cannot be translated into in vivo conditions. Therefore, it is essential to demonstrate that MSC differentiation can be manipulated and directed in vivo according to the principles of functional tissue engineering [75]. The pathway of signal transduction controlling mesenchymal stem cell differentiation seems to be more complex in vivo. Therefore, in vitro studies may not precisely mimic the signal transduction pathways controlling the development of DPSCs in vivo. The molecular mechanisms of cell differentiation in the animal or human model are partially unknown. There is a risk that after analysing the effect of a particular factor on MSC differentiation in vitro, it may omit critical interactions with complementary or competing physiologically active signalling pathways. Furthermore, the animal model may differ from predicted human responses. For therapeutic benefits, further in vivo research is needed to understand precisely the molecular mechanisms of inactivity, differentiation, self-renewal and ageing of DPSCs.

## 4. Materials and Methods

The following systematic review was conducted following the Preferred Reporting Items for Systematic Reviews and Meta-Analyses (PRISMA) guidelines [76].

In vivo studies to assess the biological response of DPSCs, their ability to differentiate into individual cell lines, the ability to migrate, proliferate and differentiate were qualified. Tests comparing the properties of DPSCs and other stem cells were accepted. Studies evaluating the influence of bioactive molecules/biomaterials/growth factors/scaffolds on DPSCs were also qualified. The electronic database search, study selection process, variable extraction and risk of bias analysis were performed by two independent researchers (T.S. and A.Z.K.). Systematic and thorough searches of the electronic database were carried out for articles published up to 7 July 2021 in PubMed-MEDLINE, Scopus, Google Scholar MEDLINE. No language or annual restrictions were applied. The article search strategy was based on the following terms: “dental pulp stem cells”, “in vivo”, “studies”, “animal model”, “human model”, “scaffold”, “growth factor”, “xenotransplantation”, “allotransplantation”, “autotransplantation”, while “AND” and “OR” were used as logical operators to combine our search terms. The search strategy and search results for the independent and linked search fields are shown in Table 1.

The inclusion criteria were articles published in English in the last ten years (from September 2011). Only full-text research manuscripts in English, German, Russian or Polish were included in this review within the last ten years. Duplicate papers were excluded. Additionally, only texts concerning DPSC in vivo studies based on human and animal models were taken into account. The quality assessment was carried following approved guidelines set by the Declaration of Helsinki, the Japanese guidelines on human stem cell clinical research and the standard for manufacturing management and the quality control of pharmaceutical products and quasi-drug (Good Manufacturing Practice, GMP). It also followed Animal Research: Reporting of In Vivo Experiments ARRIVE guidelines of the National Centre for the Replacement Refinement and Reduction of Animals in Research. Initial identification resulted in 247 titles.

## 5. Results

The selection process resulted in a total of 247 articles. We identified 242 studies through a database search (PubMed *n* = 83, Scopus *n* = 55, Google Scholar *n* = 76, Medline *n* = 28), and 5 through a hand search. After the first screening phase, duplicates, not relevant titles and articles published before 2011 were excluded. We obtained 135 titles. However, 52 papers were rejected because their full texts were not available in English or classified as abstracts or review articles. After the third selection, 43 works were excluded because they did not correspond sufficiently to the review topic, i.e., if the in vitro studies or the information about cells, growth factors and scaffolds were incomplete. After the screening procedure, according to the previously mentioned inclusion/exclusion criteria (Figure 3), 39 papers were included in the qualitative synthesis. Most of the articles demonstrate the successful use of DPSCs in tissue repair and regeneration.

The main characteristics of the studies are presented in Table 1, Table 2 and Table 3.

Depending on the donor-transplant recipient relationship, we decided to present information on xenotransplantation, allotransplantation and autotransplantation separately. The articles are categorised according to the aim of the studies, the cell source and the host. The scaffold used during the study and the growth factor were distinguished separately. The research results are shown in a separate column. In 28 studies included in this review, the cells were isolated from human tissue (xenotransplantation and human autologous transplantation). In 11 studies, the cells were isolated from different species, including dogs, rabbits, pigs, rats and mice (animal autologous transplantation and allotransplantation). The cells were transplanted into animals such as dogs, rabbits, pigs, rats and mice; an autologous human DPSC transplant was performed in four cases.

According to the clinical review of the use of dental stem cells, their multipotent properties were demonstrated, and the formation of new tissues was achieved in all types of grafts.

Animal studies showed that mesenchymal stem cells could be applied to various tissues, such as bone, the pulp-dentin complex, cementum, blood vessels, nerves, spinal cord, cartilage, skeletal muscle, pancreatic, renal and regeneration.

As far as human studies were concerned, we found four cases where the DPSCs were used—two for intrabony defect regeneration and two for pulp regeneration.

### 5.1. Cell Sources

In the case of xenotransplantation, the transplant involved cells taken from human teeth that were implanted into animals. Most human DPSCs were harvested from impacted third molars [77,78,79,80,81,82,83,84,85,86,87,88] or premolars [81,89]. The main purpose of extraction included orthodontic reasons. The age of the patients from whom the cells were isolated ranged from 6–39 years.

In the case of allotransplantation, studies were carried out mainly on rats and pigs. Regarding rats, DPSCs were obtained primarily from removed incisors [88,90,91,92,93] and molars [94]. In the case of pigs, DPSCs were isolated from molars [95]. Moreover, for the comparative study, bone marrow stromal cells were aspirated from the proximal tibia [95]. DPSCs were also isolated from incisors of mice.

In the case of the animal autologous transplantation, the cells were collected from dogs [96,97], murine and rabbits. 

Human autologous DPSC transplantation was performed in clinical trials in a total of 58 patients. In 41 cases, the tests were conducted on DPSCs isolated from deciduous teeth and in five cases on discarded teeth. The main reason for the extraction was supernumerary teeth or orthodontic purposes. In two cases, the DPSCs were collected from inflammatory pulp with diagnosed irreversible pulpitis. 

### 5.2. Scaffolds

From the analysed investigation of xenotransplantation, three groups of researchers used a cell sheet transplanted into the host to evaluate tissue regeneration characteristics [77,79,80]. Hydroxyapatite/tricalcium phosphate (HA/TCP) was used six times, Bio-Oss^®^ two times [98,99] and collagen scaffold two times [86,98]. Moreover, matrigel implants, hydrogels, gelatin sponge, alkaline phosphate (ALP), poly-L-lactic acid (PLLA), recombinant hydroxyapatite/polylactic/polyglycolic acid) (HAC/PLA/PLGA), poly-l-lactic acid/polyethylene glycol (PLLA/PEG) and 3D cell constructions were used as scaffolds to graft the DPSCs. Lee et al. [100] showed that biophysical properties of scaffolds (surface tomography, internal microstructure, scale and interconnectivity of pores/channels and material elasticity) play crucial roles in cell adhesion, migration, proliferation and differentiation in DPSCs. They created a region-specific scaffold using 3D printing with three phases of microstructures, preoptimised for the regeneration of dentin/cementum, PDL and alveolar bone from dental stem/progenitor cells. Microchannels included recombinant human amelogenin, connective tissue growth factor and bone morphogenetic protein-2. After a four-week in vivo implantation, periodontal tissues were generated. In addition, the research presented that 100 μm channels in 3D printed scaffolds are superior in promoting odontoblastic differentiation than other channel sizes tested. For animal autologous transplantation Bio-Oss, nano-hydroxyapatite/collagen/poly(l-lactide) (nHAC/PLA) and collagen were used as a scaffold. 

### 5.3. Growth Factors

For desirable tissues differentiation, the authors used an osteogenic induction medium [98,101,102,103], endothelial cell growth medium [82,90], odontogenic and cementogenic medium, adipogenic induction medium and cartilage induction medium. Some were supplemented with ascorbic acid [80,104]. In addition, in three cases, the medium features were improved by the addition of bone morphogenetic protein-2 (BMP2) [84,100,105]. For example, Atalaain et al. [84] used BMP2 to ensure the differentiation of the pulp cells into the odontoblastic lineage and induce dentin formation in vitro and in vivo. A team of researchers showed that odontogenic differentiation of the isolated and characterised human DPSCs was improved with the in vitro medium modification by the addition of BMP2. 

In research related to allotransplantation, scaffolds included dense collagen gel, self-assembling peptide hydrogel, a three-dimensional (3D) polycaprolactone (PCL)–hyaluronic acid–tricalcium phosphate (HT–PCL), SPG-178-Gel. Tsukamoto et al. [93] used an osteogenic induction medium containing recombinant human bone morphogenetic protein-4 (rhBMP-4) in a two-dimensional cell culture. After, they applied DPSCs + SPG-178-Gel, proving that it can be a suitable tool for bone formation in vivo and in vitro.

Another group of researchers led by Zhang [91] based on 33 rats with acute radioactive oesophageal injuries induced by radioactive 125I seeds, demonstrated that transplanted DPSCs could have repaired the damaged oesophageal tissue and can be an alternative approach for the treatment of the acute radioactive oesophageal injury.

For tissue differentiation, the authors applied growth factors in the form of stromal cell-derived factor-1 (SDF-1) [96], mouse-specific fibroblast growth factor-2 (FGF-2) [106], mouse-specific nerve growth factor (NGF) [106] and bone morphogenetic protein 2 (rhBMP-2)-mediated dental pulp stem cells [107]. In addition, Liu et al. [107] showed that the rhBMP-2 promoted osteogenic capability of DPSCs in New Zealand white rabbits with critical-size alveolar bone defects.

### 5.4. Osteogenic Differentiation

The nine reviewed studies presented not only DPSC application but also other stem cells, including ADSCs, BMSCs, PDLSCs, SCAPs, GMSCs, SHEDs and ABSCs [78,79,90,95,99,100,104,108,109]. Jin et al. [78] compared human ADSCs and DPSCs to repair rats’ bone defects. Although DPSCs had enhanced colony-forming ability, higher proliferative ability, stronger migration ability, higher expression of angiogenesis-related genes and secreted more vascular endothelial growth factor compared to ADSCs, ADSCs exhibited greater osteogenic differentiation potential, higher expression of osteoblast marker genes and greater mineral deposition, suggesting that ADSCs might be more useful than DPSCs for bone regeneration. A comparison of the osteogenic differentiation was also conducted by Jensen et al. [95]. The researchers investigated osteogenic differentiation of bone marrow-derived mesenchymal stromal cells (BMSCs) and dental pulp-derived stromal cells in vitro and in a pig calvarial critical-size bone defect model. The results of the in vivo study revealed that DPSCs exhibited a higher osteogenic potential compared with BMSCs.

Hu et al. [79] compared the regeneration characteristics of human cell sheets derived from dental pulp stem cells, periodontal ligament stem cells (PDLSCs) and stem cells of the apical papilla (SCAPs). In vitro studies revealed no apparent differences, whereas their regenerative characteristics in vivo were different. The tissue derived from the DPSC sheet contained much more vascular formation, and the fibres were more porous compared to the tissues derived from the other two types of sheets.

In all studies regarding DPSCs, the tissue repair and regeneration occurred compared to the control groups in which no treatment was applied.

### 5.5. Xenotransplantation

In most of the studies presented in this review, DPSCs were inserted into subcutaneous pockets prepared into the dorsal surface of mice or rats to investigate osteo/odontogenic differentiation potential [84,85,100,103,108], pulp regeneration [83] or to compare regeneration characteristics of DPSCs and PDLSCs [79] (Table 1). Moreover, the application site was also a prepared osseous defect model in mice, pigs and rabbits. In total, two studies were carried out on the cranial vault/critical size defects [99,101] and four on the alveolar/maxilla/mandibular cavities [78,80,102,105] to investigate new bone formation. Yamakakawa et al. [101] demonstrated an effective method of osteogenic differentiation of DPSCs using a helioxanthin derivative (4-(4-methoxyphenyl) pyrido [40.30: 4.5] thieno [2,3-b] pyridine-2-carboxamide (TH)) with osteogenic medium (OM). The researchers demonstrated that the transplanted DPSCs treated with OM + TH were located at the fracture sites and directly promoted bone regeneration. Similar conclusions were presented by Fujii et al. [110]. They reported that transplantation of TH-treated DPSC sheets is a convenient method of bone healing without scaffolds.

In research related to xenotransplantation, DPSCs were also inserted into the pulp chamber, periodontitis lesions, injected into a muscle, spinal cord or placed sciatic nerve crush injury model. In all these studies, the association of the stem cells with scaffolds, growth factors or recombinant proteins were common.

The eight reviewed studies presented not only DPSC application but also other stem cells, including ADSCs, BMSCs, PDLSCs, SCAPs, GMSCs, SHEDs, ABSCs [78,79,90,95,100,104,108,109]. Jin et al. [78] compared human ADSCs and DPSCs to repair rats’ bone defects. In spite of the fact that DPSCs had enhanced colony-forming ability, higher proliferative ability, stronger migration ability, higher expression of angiogenesis-related genes and secreted more vascular endothelial growth factor compared to ADSCs, ADSCs exhibited greater osteogenic differentiation potential, higher expression of osteoblast marker genes, and greater mineral deposition which suggests that ADSCs might be more useful than DPSCs for bone regeneration. Comparison of the osteogenic differentiation was also investigated by Jensen et al. [95]. Researchers investigated osteogenic differentiation of bone marrow-derived mesenchymal stromal cells and dental pulp-derived stromal cells in vitro and in a pig calvarial critical-size bone defect model. The results from the in vivo study revealed that DPSCs exhibited a higher osteogenic potential compared with BMSCs

Hu et al. [79] compared the regeneration characteristics of human cell sheets derived from dental pulp stem cells (DPSCs), periodontal ligament stem cells (PDLSCs) and stem cells of the apical papilla (SCAPs). In in vitro studies, no obvious differences were found, whereas their regenerative characteristics in vivo were different. The tissue derived from DPSC sheets contained much more vascular formation and the fibres were more porous when compared with the tissues derived from the other two types of sheets.

In all studies regarding DPSCs, tissue repair and regeneration was possible compared to the control groups in which no treatment was performed.

The research on xenotransplantation also included an investigation into the treatment of diabetic polyneuropathy through angiogenic and neurotrophic mechanisms of factors secreted by transplantation of cryopreserved human DPSCs isolated from human impacted third molars extracted for orthodontic reasons. Hata et al. [88], in previous research on hDPSCs therapeutic effectiveness, indicated that transplantation into the skeletal muscles brought about better improvement in the treatment of diabetic polyneuropathy compared to intravenous transplantation [111]. 

Ishkitiev et al. [112] established a novel culture protocol using human dental pulp stem cells (DPSCs), leading to cell generation that mimics the embryonic development of pancreatic cells.

Promising results in the application of human dental pulp stem cells (hDPSCs) and human amniotic fluid stem cells (hAFSCs) were obtained in patients with Duchenne muscular dystrophy (DMD). Previous studies showed that mesenchymal stem cells, mainly those derived from bone marrow, have potential use in cell therapy for DMD-related diseases or other muscular dystrophies [76]. Pisciotta et al. [113] showed that by intramuscular injection of hDPSCs and hAFSCs into mice representing an animal model of DMD, an improvement in the histopathology of the dystrophic muscle was obtained. Furthermore, regeneration took place through the paracrine effect, promoting angiogenesis and reduction of fibrosis [87]. The relationship between promoting angiogenesis and improving muscle function was demonstrated previously by Palladino et al. [114] and Verma et al. [115].

Regenerative properties based on angiogenesis were also investigated by Angelopoulos et al. [82] They compared the biological and functional properties between DPSC and GMSC. The ability to induce angiogenesis was also presented by Bronckaers et al. [116] and Hilkens et al. [117].

Based on in vitro and in vivo studies, Angelopoulos et al. demonstrated that GMSCs have a higher ability to proliferate, migrate and create angiogenic tubules compared to DPSC [82]. The detailed mechanism of signalling events regulating the vasculogenic fate of dental pulp stem cells was presented by Zhang et al. [90]. 

Yang et al. used a rat model of a completely transected spinal cord to indicate neurite regeneration after spinal cord injury. The previous investigation demonstrated that dental stem cells can secret neurotrophic factors and promote the migration and survival of neurons [118]. Yang et al. demonstrated that dental stem cells differentiated into mature neurons and oligodendrocytes but not astrocytes, reduced inflammatory response, progressive haemorrhagic necrosis and promoted neurite regeneration. 

To stimulate the regeneration of cartilage lesions, the use of chondrocyte grafts is not always possible [119]. As an alternative, Rizk et al. introduced DPSCs and demonstrated that if seeded on a PLLA/PEG electrospun scaffold, they could form in vivo three-dimensional cartilage constructs. However, further research is needed to verify that DPSCs can maintain the differentiation potential after transplantation [120].

The main directions in research on xenotransplantation were the investigation of regeneration dental-pulp complex [79,81,86,102,103]. In all studies, it was shown that DPSCs stimulate pulp recovery and dentin regeneration in comparison to control groups. Zhai et al. [102] investigated the expression and biological function of human β-defensin 4 (HBD4) in DPSCs and explored its potential as a pulp capping agent using a rat model of reversible pulpitis. DPSCs and HBD4 down-regulated the expression of inflammatory mediators, enhanced the differentiation of DPSCs into osteoblasts or odontoblasts and induced the formation of restorative dentin. Even though a decrease in the functionality of stem cells can be observed [121] and reduced regenerative capacity [122] with age in most tissues, Horibe and colleagues [86] showed that MDPSC regenerative abilities are age-independent. Among the examined properties, they assessed the migration and proliferation potential, trophic and anti-apoptotic effect and expression of angiogenic/neurotrophic factors. They indicated that in the hindlimb model and an ectopic tooth root model, the properties of aged MDPSCs and young MDPSCs were similar [86].

Dental tissue regeneration using DPSCs and PDLSC was demonstrated by Cha et al. [108] based on in vivo and in vitro studies. They examined the regeneration effects from pre-differentiated DPSCs and PDLSCs at different periods of time. No significant increase in tissue-forming ability was observed; however, DPSCs generated hard tissue closer to dentin.

### 5.6. Allotransplantation

One of the research directions was the use of DPSCs in the regeneration of bone defects. In three cases, the research was carried out on calvarial defects. This reference animal model for testing the effectiveness of osteoconductive substitutes is common and reliable and ensures minimal morbidity and perioperative mortality [123]. In all cases, the bone mineral density and bond protector parameters were significantly increased. In Jensen’s [95] study of 28 pigs, it was confirmed that DPSCs showed superior bone healing potential compared to BMSCs both in vitro and in a large-animal bone defect. Furthermore, the researchers presented that a polycaprolactone (PCL)–hyaluronic acid–tricalcium phosphate (HT–PCL) scaffold showed good osteoconductive properties compared to both an empty control and a polycaprolactone PCL scaffold without surface modification. Previous clinical studies showed that an appropriate scaffolding in combination with MSCs could promote mineralisation and healing of large bone loss through increases in collagen fibrillary density [124]. The clinical use of mesenchymal stem cells derived from the dental pulp was investigated for diabetic treatment by two groups of researchers. In Hata’s [125] experiments, transplantation of DPSCs significantly improved the impaired sciatic nerve blood flow, sciatic motor/sensory nerve conduction velocity, capillary number to muscle fibre ratio and intra-epidermal nerve fibre density in the transplanted side of diabetic rats. Research shows that it may be an effective method for the treatment of diabetic neuropathy. Evaluations of the therapeutic potential of DPSCs in diabetes were also presented by Guimarães et al. [126]. They investigated endovenous transplantation of DPSCs isolated from the incisor teeth of male EGFP transgenic mice and their impact on pancreatic damage, renal function alterations and diabetic peripheral neuropathy. The researchers suggested stem cell therapy as an option for the control of diabetes complications. The above studies were confirmed by Omi et al. [127], who demonstrated that the use of DPSCs stimulated the neurite outgrowth of dorsal root ganglion neurons and increased the viability and myelin-related protein expression of Schwann cells in both small nerve fibres to large nerve fibre, also in long-term diabetic animals. DPSCs may contribute to accelerating the regeneration of the renal tubules, which was presented by Barros et al. [128] based on the ARF rat model. In the case of allotransplantation, the improvement in kidney function after the transplantation of mDPSCs in the mice model was obtained in the studies by Guimarães et al. [126] The results showed that urea and proteinuria levels normalised, and kidney function improvement was also confirmed by the histopathological analysis of the kidney sections.

Intravenous transplantation was also conducted by Zhang et al. [92], regarding the therapeutic effects of DPSCs for ischaemic vascular diseases and observed neuroprotective effect in brain ischaemia rats by reducing the infarct volume and enhancing the neurological function recovery after cerebral ischaemic injury. Neuroprotective properties were also investigated by Mead et al. [129] on axotomised adult rat retinal ganglion cells (RGC) in vitro. Compared to rat BMSC, rat DPSC promoted substantially higher neurotrophin-mediated neuroprotection and axon regeneration both in vitro and in vivo.

### 5.7. Autotransplantation

Autotransplantation studies were carried out on animal models (Table 3A) and clinical trials with humans (Table 3B). In Khorsand’s [97] research conducted on 20 dogs from which DPSCs were isolated from premolar maxillary teeth, periodontal regeneration of DPSCs was observed. Transplantation of DPSCs was combined with Bio-Oss granules. The results obtained regeneration of bone, periodontal ligament (PDL) and cement. Woven bone was obtained in a rat model compared to other studies conducted with DPSCs combined with collagen [130]. Based on the canine model pulp regeneration capacity of DPSCs was also confirmed.

Iohara et al. [96] demonstrated complete pulp regeneration with neurogenesis and vasculogenesis in dogs after pulpectomy in mature teeth. Autotransplantation of pulp progenitor cells (CD105 (+)) was performed with a stromal cell-derived factor-1 (SDF-1). Collagen scaffold was used in the research due to its excellent biocompatibility. It was shown that new pulp-like tissue with nerves and vessels appeared in the canals. Newly formed dentin was also shown on the canal walls, and there was complete apical closure.

Neural therapeutic transplantation in pre-clinical studies was also shown in a mouse model of autotransplantation. Ellis et al. [106] reported the in vitro immature neuronal-like networks development of DPSC isolated from murine incisors using a neural differentiation methodology, with a mouse-specific fibroblast growth factor-2 (FGF-2) and mouse-specific nerve growth factor (NGF). The researchers showed that DPSCs could be used in the treatment of neurodegenerative diseases. Additionally, in another study by Sakai et al. [10], the regenerative abilities of mesenchymal cells were demonstrated, especially in stimulating the regeneration of damaged axons. The data presented by the researchers may be useful for further preclinical investigation of the neurological recovery following stroke and traumatic injury.

Li et al. [131] research showed that a DPSC autograft transplanted with a β-tricalcium phosphate scaffold β-TCP complex dramatically improved the clinical symptoms of periodontitis when comparing the biological characteristics of normal DPSCs and those isolated from inflammatory dental tissues. As the first, they confirmed the decreased osteogenic ability by autologous DPSCs, despite the teeth inflammatory pulp tissue, and can be a suitable cell source for tissue regeneration, especially in intrabone defect in periodontitis.

The regenerative potential of autologous DPSCs in the treatment of human uncontained intraosseous defects was also proven by Aimetti et al. [132]. At the 1-year examination, a chronic periodontitis patient was surgically treated. His intrabony defect on the mandibular right second premolar was filled with bone-like tissue, as confirmed through the re-entry procedure. The scientists achieved a decrease of gingival recession defects in 11 healthy patients. 

The regenerative potential of DPSCs in autologous human transplantation was shown in the treatment of irreversible pulpitis [39] and pulp necrosis after traumatic dental injuries [133]. The American Association of Endodontists (AAE) defines regenerative endodontics as ‘‘biologically based procedures designed to replace damaged structures, including dentin and root structures, as well as cells of the pulp dentin complex’’ [134].

In experiments of Nakashima [39], five patients with irreversible pulpitis were enrolled and monitored for up to 24 weeks following mobilised dental pulp stem cell transplantation from discarded teeth. Granulocyte colony-stimulating factor (G-CSF) was used as a growth factor, and atelocollagen was applied as a scaffold. A pilot clinical study demonstrated the therapeutic potential of MDPSCs for complete pulp regeneration suggested by EPT, MRI and cone-beam computed tomography. In a randomised controlled clinical trial, the regrowth of destroyed pulp tissue was also obtained in Xuan et al. [133]. They enrolled 40 patients with pulp necrosis after traumatic dental injuries. DPSCs were implanted into injured incisor teeth in the absence of a scaffold and promoted three-dimensional pulp regeneration, containing normal structures such as an odontoblast layer, connective tissue, blood vessels and nerves after 12 months since treatment. Comparing the apexification control group, the autologous DPSC transplant group showed an increase in root length and reduction in the width of the apical foramen 12 months after treatment.

**Table 1 molecules-26-07423-t001:** Xenotransplantation.

Xenotransplantation (From Human to Animal)							
Case No.	Aim	Cell Source	Host	Scaffold/Cell Sheet	Growth Factor	Results	Article
1.	To explore the potential roles and molecular mechanisms of DPSCs in crushed nerve recovery.	human DPSCs extracted third molars or orthodontic teeth (15–25 years)	32 adult male SD rats had nerve crush injury	cell sheets and N-DPSC	epidermal growth factor basic fibroblast growth factor	DPSCs are inclined to differentiate into neural cells. Could help crushed nerves with functional recovery and anatomical repair in vivo. Thus, DPSCs or N-DPSCs could be a promising therapeutic cell source for peripheral nerve repair and regeneration	[77]
2.	Comparison of the bone formation capacity of DPSCs and ADSCs in vitro and in vivo.	hDPSCS (third molars) from 20–25-year-old individuals; ADSCs from 25–35-year-olds during liposuction.	15 rats mandibular bone defect	alkaline phosphate (ALP)		Indicated the extensive potential of the DPSCs in tissue repair and regeneration. ADSCs exhibited greater osteogenic differentiation potential, higher expression of osteoblast marker genes and greater mineral deposition	[78]
3.	Comparison of the regeneration characteristics of cell sheets derived from dental pulp stem cells (DPSCs), periodontal ligament stem cells (PDLSCs) and stem cells of the apical papilla (SCAPs).	Human (DPSC, PDLSCs) and (SCAPs)—impacted third molars	subcutaneously into the dorsal surfaces of 5 10-week-old mice	cell sheet	vitamin C, hydroxyapatite/tricalcium phosphate (HA/TCP)	Although in vitro DPSC, PDLSC and SCAP cell sheets have similar characteristics, their regenerative characteristics in vivo are different, with each showing potential application for regeneration of different tissues. Dental pulp stem cell sheet formed a loose connective tissue, rich in blood vessels, similar to dental pulp tissue, suggesting that DPSC sheet could be more suitable for dental pulp or vascular rich tissue regeneration.	[79]
4.	Evaluation the effect of cell injection and cell sheet transplantation on periodontal regeneration in a swine model.	human DPSCs	12 pigs were used to generate periodontitis lesions of the first molars for a total of 24 defects	HDPSC sheet group hDPSC injection group	Vc xenobiotic-free cell culture reagents	Xenogeneic DPSC sheets and DPSC injection can be appropriate therapies for periodontal bone and soft tissue regeneration	[80]
5.	DPSCs and human umbilical vein endothelial cells (HUVECs) were used to evaluate the biological effects of SAP-based scaffolds.	hDPSC premolars, third molars (18–25 years)	35 rats	Extracellular matrix (ECM)-like biomimetic hydrogels composed of self-assembling peptides (SAPs) scaffold with SAPs	Morphogenic signals in the form of growth factors (GFs)	DPSCs grown on this composite scaffold stimulating pulp recovery and dentin regeneration in vivo	[81]
6.	Identification of the optimal dental source of MSCs through a biological and functional comparison of gingival (GMSCs) and dental pulp stem cells (DPSCs) focusing mainly on their angiogenic potential	human MSCs from the dental pulp of the third molars and gingival tissues of the same patient	24 NSG mice	Matrigel implants	endothelial cell growth medium	GMSCs displayed a higher capacity to proliferate, migrate and form angiogenic tubules compared with DPSCs in vitro and in vivo	[82]
7.	Assessment viability of these 3D DPSC constructs for dental pulp regeneration through in vitro and in vivo studies	DPSCs from human adult third molars	DPSC were inserted into the human root canal, and then transplanted into the subcutaneous space of 6 mice	Rod-shaped 3D cell construct	OM for odontoblastic differentiation	DPSC constructs possess self-organizing ability and can be used for novel dental pulp regeneration therapy; fabrication of a scaffold-free, rod-shaped cell construct composed of DPSCs, using thermoresponsive hydrogel	[83]
8.	Whether medium modification improves the odontogenic differentiation of human dental pulp stem cells (DPSC) in vitro and in vivo	DPSC human impacted third molar teeth	subcutaneous dorsal surface of the mice	hydroxyapatite tricalcium phosphate scaffold	bone morphogenetic protein 2 (BMP2)	Odontogenic differentiation of the isolated and characterised human DPSC was improved with medium modification by the addition of BMP2 in vitro and in vivo	[84]
9.	Peptide hydrogel PuraMatrix™ was used as a scaffold system to investigate the role of dental pulp stem cells (DPSCs) in triggering angiogenesis and the potential for regenerating vascularised pulp in vivo	DPSCs from extracted sound third molars from humans (18 to 25 years)	Root segments were implanted in the subcutaneous space of the dorsum of 20 5- to 7-week-old mice	peptide hydrogel PuraMatrix™	-	Importance of a microenvironment that supports cell–cell interactions and cell migration, which contribute to successful dental pulp regeneration	[85]
10.	Comparison of the biological properties of aged MDPSCs versus young MDPSCs	DPSCs from human third molars were collected from younger (19–30 years, *n* = 6) and older (44–70 years, *n* = 6)	SCID Mice (ischemic hidlimb) SCID Mice (subcutaneous)	Tooth roots, collagen TE	-	The regenerative potential of MDPSCs is independent of age, demonstrating an immense utility for clinical applications by autologous cell transplantation in dental pulp regeneration and ischemic diseases	[86]
11.	Investigation of the potential of human dental pulp stem cells (hDPSCs) and human amniotic fluid stem cells (hAFSCs) to differentiate toward a skeletal myogenic lineage using several different protocols	human DPSCs (enclosed third molar of teenage subjects) hAFSCs	mdx/SCID mice (gastrocnemius muscles (GMs)	Intramuscular injection of pre-differentiated DPSCs in myogenic medium	-	Promoted angiogenesis and reduced fibrosis, improvement of pathological features of dystrophic skeletal muscle tissues, regeneration of muscles in Duchenne muscular dystrophy	[87]
12.	Investigation of the therapeutic potential of intravenous and Intrapancreatic transplantation of human dental pulp stem cells in a rat model of streptozotocin-induced type 1 diabetes	DPSCs from human impacted third molars	40 rats	hDPSCs were injected into the pancreas or tail vein after the induction of diabetes in nude mice	-	Human dental pulp stem cells can migrate and survive within streptozotocin-injured pancreas and induce antidiabetic effects through the differentiation and replacement of lost β-cells and paracrine-mediated pancreatic regeneration	[88]
13.	Engineering sizable three-dimensional cartilage-like constructs using stem cells isolated from human dental pulp stem cells (DPSCs)	DPSCs from human premolars extracted for orthodontic treatment	10 mice (8–10 weeks)	poly-l-lactic acid/polyethylene glycol (PLLA/PEG) electrospun fiber scaffolds	growth factor β3 (TGFβ3)	Immuno-selected DPSCs can be successfully differentiated toward chondrogenic lineage; it may be useful in future treatment of cartilage defects	[89]
14.	How human mesenchymal stem cells differentiate after birth into endothelial cells that make up blood vessels	human permanent teeth (DPSC) or deciduous teeth (SHED)	MSCs seeded in human tooth slice/scaffolds were transplanted 8 mice	poly-L-lactic acid (PLLA) scaffold	vasculogenic differentiation medium, i.e., endothelial cell growth medium (EGM2-MV, Lonza) supplemented with rhVEGF165.	VEGF signalling through the canonical Wnt/β-catenin pathway defines the vasculogenic fate of postnatal mesenchymal stem cells dental pulp stem cells can differentiate into endothelial cells that form blood vessels	[90]
15.	Illuminate the role of hsa_circ_0026827 in human dental pulp stem cells (DPSCs) during osteoblast differentiation.	human DPSCs	15 mice	Bio-Oss Collagen scaffolds	osteogenic medium	hsa_circ_0026827 promotes osteoblast differentiation of DPSCs	[98]
16.	Investigation of whether the combination of Bio-Oss scaffold with BMSCs and DMSCs promotes improved bone regeneration and osteogenesis-related protein expression in a rabbit calvarial defect model	human DPSCs and BMSCs	Rabbit calvarial defects	xenografts bio-oss		In the in vivo studies, the bone volume density in DPSCs group was significantly greater than that in the empty control or Bio-Oss only group	[99]
17.	Comparison of multiphase region-specific microscaffolds (polycarprolactione-hydroxylapatite) with spatiotemporal delivery of bioactive cues for integrated periodontium regeneration.	human DPSCs, PDLSCs, and ABSCs from 18–39-year-old patients	20 mice (dorsum’s midsagittal plane)	Polycarprolactione-hydroxylapatite (90:10 wt%) scaffolds	GF Recombinant human amelogenin, connective tissue growth factor, and bone morphogenetic protein-2	DPSC appears to differentiate into putative dentin/cementum, PDL and alveolar bone complex by scaffold’s biophysical properties and spatially released bioactive cues	[100]
18.	To investigate the localisation of transplanted DPSCs in a mouse fracture model	human DPSCs	27 mice (calvarial defect model)	-	helioxanthin derivative 4-(4-methoxyphenyl)pyrido[40,30:4,5]thieno[2,3-b]pyridine-2-carboxamide (TH)) and osteogenic medium	OM + TH-treated DPSCs promoted fracture healing. Moreover, transplanted DPSCs had localised to the fracture site and were directly involved in fracture healing.	[101]
19.	Investigation the expression and biological function of human β-defensin 4 (HBD4) in dental pulp stem cells (DPSC) and explored its potential as a pulp capping agent	human DPSCs	15 8-week-old male Wistar rats (holes in the centre of the bilateral maxillary first molar surface to expose the pulp chamber)	gelatin sponge	osteogenic induction medium adipogenic induction medium cartilage induction medium	DPSC (with expression and biological function of human β-defensin 4 HBD4) controlled the degree of pulp inflammation in a rat model of reversible pulpitis and induced the formation of restorative dentin. DPSC may be a useful pulp capping agent for use in vital pulp therapy VPT.	[102]
20.	Comparison of the stemness and differentiation potential of ACCs and DPSCs of human immature permanent teeth with the aim of determining a more suitable source of stem cells for regeneration of the dentin-pulp complex	human DPSCs 13 from permanent teeth of 12 children aged 6–18 years	15 mice subcutaneous pockets made in 5-week-old male	biphasic calcium phosphate	osteogenic medium	In the in vivo study, ACCs and DPSCs formed amorphous hard tissue using macroporous biphasic calcium phosphate particles. Regarding regeneration of the dentin-pulp complex, the coronal pulp can be a suitable source of stem cells considering its homogenous lineages of cells and favorable osteo/odontogenic differentiation potential.	[103]
21.	Exploration of the survival, differentiation and immunomodulatory ability of transplanted cells in the extreme inflammatory environment, and to investigate tissue regenerative capability and possible corresponding mechanisms of transplanted cells after spinal cord injury	human (18–22 years) DPSCs SHEDs	32 male Wistar rats (10th spinal cord was completely transected)	natural and artificial scaffold	medium with ascorbic acid	DFSC demonstrated the potential in repairing the completely transected spinal cord and promoting functional recovery after injury	[104]
22.	Evaluation clinical, histological and radiological osseous regeneration in a critical-sized bilateral cortico-medullary osseous defect in model rabbits from New Zealand after receiving a hydroxyapatite matrix and polylactic polyglycolic acid (HA/PLGA) implanted with human dental pulp stem cells (DPSCs)	human DPSCs extracted teeth for orthodontic reasons	8 rabbits with critical-sized bilateral cortico-medullary osseous defect	hydroxyapatite matrix and polylactic polyglycolic acid (HA/PLGA)/DPSC matrix	BMP	HA/PLGA/DPSC scaffold was an effective in vivo method for mandibular bone regeneration	[105]
23.	Determination of the effects of in vitro odontogenic/cementogenic differentiation on the in vivo tissue regeneration of (DPSCs) and (PDLSCs)	human DPSC from 16 human teeth and PDLSCs	subcutaneously transplanted into the dorsal surface of 5-week-old male mice (*n* = 45)	scaffold macroporous biphasic calcium phosphate	odontogenic/cementogenic medium	Predifferentiated DPSCs and PDLSCS generated hard tissue closer to dentin and higher-quality and greater amounts of tissue for dental regeneration than undifferentiated	[108]
24.	Differentiation of SHED and DPSCs into islet cells and assessment of their insulin secretory capacity in vitro and in vivo	SHED and DPSCs were obtained from human teeth (5–40 years old)	Balb/C 40 male mice, 6–8 weeks old	immuno-isolatory biocompatible macro-capsules	polyurethane-polyvinylpyrrolidone semiinterpenetrating network	Differentiation DPSCs to islet cells aggregates (ICA) similar to pancreatic islet cells. T source of human tissue that could be used for management of diabetes type 1.	[109]

**Table 2 molecules-26-07423-t002:** Allotransplantation.

Allotransplantation (Animal Transplantation within the Same Species)							
Case No.	Aim	Cell Source	Host	Scaffold/Cell Sheet	Growth Factor	Results	Article
1.	Exploration of the therapeutic effects of DPSCs on acute radiation-induced oesophageal injury	rat DPSC isolated from the incisors	33 rats with acute radioactive oesophageal injuries induced by radioactive 125I seeds in vivo		The OriCellTM osteogenesis differentiation kit was used to induce osteogenic differentiation. P3 SCs were cultured in osteogenic differentiation medium Adipogenesis was induced by using the OriCellTM adipogenic differentiation kit (injected with DPSCs (1 × 10^7^ cells)	The results demonstrated that transplanted DPSCs, which trans-differentiated into esophageal stem cells in vivo, could repair the damaged esophageal tissue	[91]
2.	Investigation of the therapeutic potential of DPSCs for ischemic vascular diseases and opportunity for neural regeneration	DPSCs was harvested from the incisors of 4-week-old male SD rats	24 rats with middle cerebral artery occlusion (MCAO)	Intravenous infusion of DPSCs		Neuroprotective effect on brain ischemia rats, by reducing the infarct volume and enhancing the neurological function recovery after cerebral ischemic injury	[92]
3.	Verification of DPSCs proliferation and osteogenic differentiation in a three-dimensional cell culture using SPG-178-Gel	DPSCs isolated from the dental pulp of extracted incisors of six-week-old male Sprague-Dawley (SD) rats	24 h-week-old male Sprague-Dawley (SD) rats with the calvarial defect	self-assembling peptide hydrogel, SPG-178-Gel,	Osteogenic induction medium containing recombinant human bone morphogenetic protein-4 (rhBMP-4) in a two-dimensional cell culture	In conclusion, DPSCs + SPG-178-Gel can be a suitable tool for bone formation in vivo and in vitro	[93]
4.	Evaluation of the osteogenic effects of dense collagen gel scaffolds seeded with rat DPSC (rDPSC) implanted in a rat critical-sized calvarial defect model	DPSC isolated from the molars of 4-day Wistar rats	30 rats with critical-size calvarial defect model	dense collagen gel scaffolds		Bone mineral density and bone micro-architectural parameters were significantly increased when DPSC-seeded scaffolds were used	[94]
5.	Comparison of the osteogenic differentiation of bone marrow-derived mesenchymal stromal cells and dental pulp-derived stromal cells (DPSCs) in vitro and in a pig calvaria critical-size bone defect model	DPSCs isolated from the premolar from the upper and lower pig jaw, BMSCs was aspirated from the proximal tibia	28 pigs with critical-size calvarial defect model	a three-dimensional (3D) polycaprolactone (PCL)–hyaluronic acid–tricalcium phosphate (HT–PCL) scaffold.		DPSCs exhibited a higher osteogenic potential compared with BMSCs both in vitro and in vivo, making it a potential cell source for future bone tissue engineering	[95]
6.	Assessment of the therapeutic potential of DPSCs transplant in the case of diabetic polyneuropathy	DPSCs isolated from the dental pulp of extracted incisors of Sprague-Dawley rats	10 points of normal and diabetic rats			Transplantation of DPSCs could be a promising tool for the treatment of diabetic neuropathy	[125]
7.	Evaluation of the therapeutic potential of mDPSCs in important complications of diabetes, namely pancreatic damage, renal function alterations and diabetic peripheral neuropathy	DPSCs DPSCs isolated from the incisor teeth of male EGFP transgenic C57BL/6 mice	12 diabetic mice	Endovenous transplantation		Improved pancreatic damage, renal function and painful neuropathy	[126]

**Table 3 molecules-26-07423-t003:** (**A**) Autotransplantation (animal autologous transplantation). (**B**) Autotransplantation (human autologous transplantation).

**(A)**							
**Autotransplantation (Animal Autologous Transplantation)**							
**Case No.**	**Aim**	**Cell Source**	**Host**	**Scaffold/Cell Sheet**	**Growth Factor**	**Results**	**Article**
1.	The demonstration for the first time of complete pulp regeneration in the root canal, after pulpectomy, in dogs	Dog CD105+ DPSCs	5 dogs after pulpectomy in mature teeth	a collagen scaffold	stromal cell-derived factor-1 (SDF-1)	Complete pulp regeneration with neurogenesis and vasculogenesis occurred	[96]
2.	Evaluation of effects of dental pulp stem cells (DPSCs) on regeneration of a defect experimentally created in the periodontium of a canine model	DPSCs isolated from 2 maxillary premolar dog teeth	20 dogs	Bio-Oss		DPSCs are capable of promoting periodontal regeneration	[97]
3.	Demonstration of the neuronal differentiation of DPSC from murine incisors	DPSCs isolated from murine incisor teeth	murine		mouse-specific fibroblast growth factor-2 (FGF-2) mouse-specific nerve growth factor (NGF)	Generated neuronal-like cells from murine incisor DPSC to an immature stage of development. Our findings encourage the use of mDPSC to develop mouse models of autologous neural therapeutic transplantations for pre-clinical studies.	[106]
4.	Evaluation of the capacity of a Tissue-engineered bone complex of recombinant human bone morphogenetic protein 2 (rhBMP-2)-mediated rabbit dental pulp stem cells (DPSCs) and nano-hydroxyapatite/collagen/poly(L-lactide) (nHAC/PLA) to reconstruct critical-size alveolar bone defects in New Zealand rabbit	DPSCs from New Zealand white rabbit	36 rabbits with critical-size alveolar bone defects	scaff-nano-hydroxyapatite/collagen/poly(L-lactide) (nHAC/PLA)	bone morphogenetic protein 2 (rhBMP-2)-mediated dental pulp stem cells (DPSCs)	The rhBMP-2 promoted osteogenic capability of DPSCs as a potential cell source for periodontal bone regeneration. DPSCs might be a better alternative to autologous bone for the clinical reconstruction of periodontal bone defects.	[107]
**(B)**							
**Autotransplantation (Human Autologous Transplantation)**							
**Case No.**	**Aim**	**Cell Source**	**Host**	**Scaffold/Cell Sheet**	**Growth Factor**	**Results**	**Article**
1.	Evaluation of the safety, potential efficacy and feasibility of autologous transplantation of MDPSCs in pulpectomised teeth	DPSCs isolated from discarded teeth	5 patients with irreversible pulpitis	atelocollagen	granulocyte colony-stimulating factor (G-CSF)	Pulp regeneration, functional dentin formation in three of the five patients	[39]
2.	Trying to isolate of DPSCs-IPs from two patients and to evaluate the feasibility and the effect of reconstructing periodontal intrabone defects in each patient	DPSCs-IPs derived from inflammatory dental pulp tissues	2 patientswith cells engrafted into the periodontal defect area in the root furcation.	β-tricalcium phosphate (β-TPC	-	Regeneration of new intrabony defect	[131]
3.	The description of the clinical and radiographic regenerative potential of autologous DPSCs in the treatment of human uncontained intraosseous defects	DPSCs collected from deciduous teeth	1 patient		-	The defect was completely filled with bonelike tissue	[132]
4.	To show that implantation of autologous tooth stem cells from deciduous teeth regenerated dental pulp with an odontoblast layer, blood vessels and nerves	DPSCs isolated from deciduous teeth	40 patients with pulp necrosis after traumatic dental injuries	extracellular matrix	-	Regeneration of dental pulp tissue containing sensory nerves	[133]

## 6. Conclusions

Recent studies have shown that the dental pulp contains multipotent mesenchymal stem cells capable of differentiating into multiple lines under the right conditions. Bone marrow (BM) is considered the main source of MSCs [135]; however, obtaining BM from a patient is painful and invasive. Therefore, there is a need to find other alternative tissue sources that are non-invasive and produce similar MSC types. Several researchers are currently studying adipose-derived stem cells, human amniotic fluid mesenchymal stem cells (AFMSC) [136], menstrual stem cells (MenSCs) [137], umbilical cord blood stem cells [138], synovial fluid mesenchymal stem cells (SFMSC) [139] and DPSCs for this purpose. Now, it has been established that all of these cells have similar BM cell renewal and differentiation properties. DPSCs are a promising source of cells and can be used for future regenerative therapies for various diseases; however, multidisciplinary collaboration between biomedical scientists and engineers is needed to identify the mechanisms of cellular interactions and fully exploit their regenerative abilities in living organisms.

The combination of growth factors, small molecules, scaffold materials and optimal culture conditions are crucial for the proper differentiation of DPSCs and unlocking their regenerative potential. In our systematic review, we analysed in vivo human and animal studies using xenotransplantation, allotransplantation and autotransplantation. The usefulness and advantages of DPSCs, such as the ability to self-regenerate and multi-directional differentiation, easy access and, notably, the stability of regenerated tissues, were confirmed. To optimise the effects of treatment with mesenchymal stem cells, it is necessary to monitor their activity after transplantation. In vivo animal model studies are important for adjusting the treatment regimen in human clinical trials. The literature review shows that materials engineering and molecular biology in regenerative medicine allow the reconstruction of damaged tissues and organs. Further controlled clinical trials are needed to understand the regeneration process fully. It has been shown that DPSCs, thanks to their plasticity, can form a dentin-pulp complex, nerves, fat tissue, bone, cartilage, skin, blood vessels and the heart muscle. This makes pulp stem cells a promising tool in treating a very diverse group of human diseases.

## Figures and Tables

**Figure 1 molecules-26-07423-f001:**
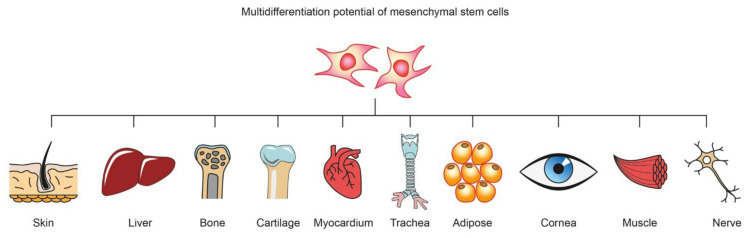
Graphical representation of clinical use of Mesenchymal Stem Cells.

**Figure 2 molecules-26-07423-f002:**
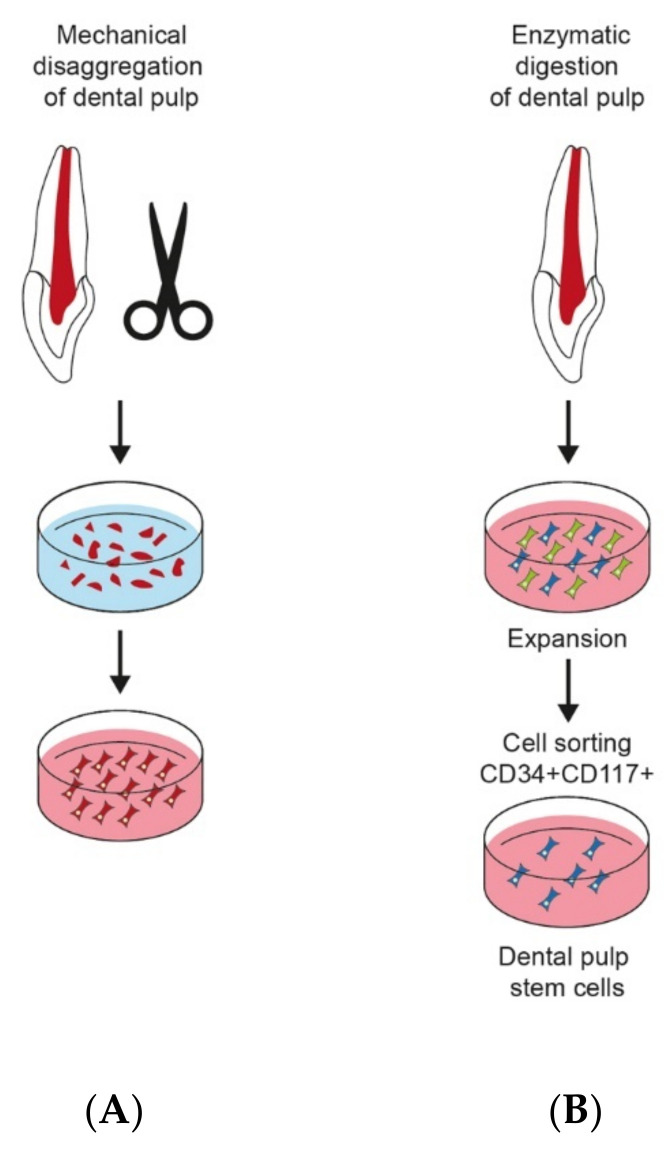
Main methods to culture dental pulp stem cells: (**A**) Explant method—dental pulp is fragmented into pieces and cultured in medium; (**B**) enzymatic digestion method—dental pulp is digested, and suspension is screened for expression markers by flow cytometry.

**Figure 3 molecules-26-07423-f003:**
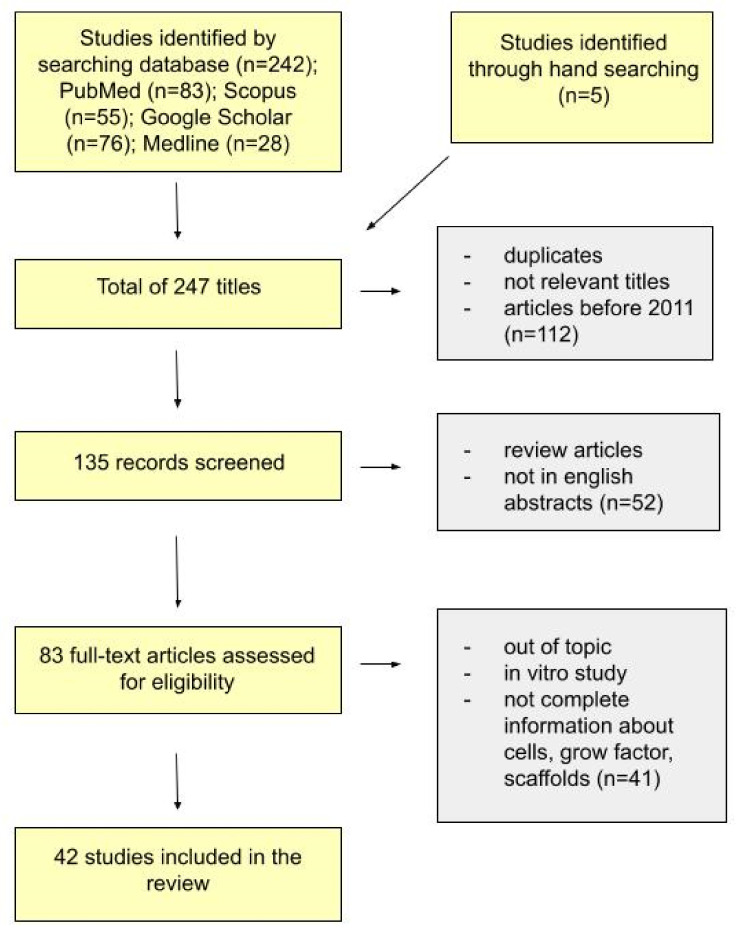
PRISMA flow diagram of article selection in this systematic review.

## Data Availability

Not applicable.

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
