# Peer review of "Therapeutic Potential of Dental Pulp Stem Cells According to Different Transplant Types"

_molecules, 2021, doi:10.3390/molecules26247423_

Round 1

Reviewer 1 Report

Overall, the manuscript is well written and covered several aspects regarding dental pulp stem cells. However there are a few points that need to be addressed;

Section 2.1 title is misleading, it should be surface markers instead of molecular markers.

Section 2.4. Methods of Stem Cells Characterization; needs more clarification. How is Quantitative real-time polymerase chain reaction (qRT-PCR) used in characterisation? Also, list the -flow cytometry based on fluorochrome-conjugated antibodies used. Details are needed in this section as a whole.

Author Response

Dear Editor,

Thank you very much for the review of our Manuscript IDmolecules-1478891.

We sincerely appreciate all valuable comments and suggestions, which helped us to improve the quality of the article. Our responses to the Reviewer comment are described below in a point-to-point manner.

 Appropriated changes, suggested by the Reviewers, has been introduced to the manuscript such as change of title, details in some paragraphs etc. Moreover, several authors, along with her affiliation and contribution to the work, have been added.

Section 2.1 title is misleading, it should be surface markers instead of molecular markers.

Suggestion accepted. This sentence has been changed.

Section 2.4. Methods of Stem Cells Characterization; needs more clarification. How is Quantitative real-time polymerase chain reaction (qRT-PCR) used in characterisation? Also, list the -flow cytometry based on fluorochrome-conjugated antibodies used. Details are needed in this section as a whole.

Suggestion accepted.

Section 2.4 has been expanded

QRT-PCR

This method is used to duplicate the DNA fragments. With the help of the polymerase chain reaction (PCR technique), we can get billions of copies of a specific DNA sequence in a short time. Then we can determine its length and exact structure. Quantitative real-time polymerase chain reaction (qRT-PCR)  method enables the measurement of the number of new copies of the reaction product at any point in time (in real time). It is a very sensitive technique that enables the detection of even one copy of the studied gene. It differs from traditional PCR in that it analyzes the increase of PCR reaction products after each reaction cycle, not only after the last one. Due to its specificity, the qPCR method has become a valuable diagnostic tool. By this method we can identified numerous human MSC-characteristic genes.

Kubo H, Shimizu M, Taya Y, Kawamoto T, Michida M, Kaneko E, Igarashi A, Nishimura M, Segoshi K, Shimazu Y, Tsuji K, Aoba T, Kato Y. Identification of mesenchymal stem cell (MSC)-transcription factors by microarray and knockdown analyses, and signature molecule-marked MSC in bone marrow by immunohistochemistry. Genes Cells. 2009 Mar;14(3):407-24. doi: 10.1111/j.1365-2443.2009.01281.x. Epub 2009 Feb 17. PMID: 19228201.

Flow cytometry

Flow cytometry enables the analysis of several parameters for each cell, cell fragments, or other artifacts as the color intensity, size, and fluorescence of the tested cells and. The results of the diagnosis are histograms which allow to visualize the populations of cells with specific characteristics in the tested sample. Cytometers equipped with a device for sorting cells in an electric field make it possible to evaluate some functional parameters of cells and isolation appropriate cells.

Nakashima M, Iohara K. Regeneration of dental pulp by stem cells. Adv Dent Res. 2011 Jul;23(3):313-9. doi: 10.1177/0022034511405323. PMID: 21677085.

We truly hope that the revised manuscript is now clarified enough for the reviewer to take the next step. We deeply appreciate your help to improve the readability of our paper.

Yours sincerely,

Anna Zawadzka- Knefel

Reviewer 2 Report

  1. The given title is misleading. Authors should focus on the in vivo therapeutic potential of DPSCs. However, they discussed a lot about the in vitro part?!!
  2. Line 229. Authors mentioned that “The same method was used to isolate DPCs and DPSCs to 228 compare gene expression with MSCs and FBS [52]”

Which MSCs? I understand that FBS referred to Fetal Bovine Serum as you abbreviated elsewhere however the authors compare with fibroblasts??!!!

3.Line 82. In fact, not only ADMSC and DPSC that are currently studies but other many sources, as you mentioned above, such as AFMSC, SFMSC, endometrial, umbilical…etc.

  1. Line 128, not only Gronthos who did the comparison but with other authors as well, “he” should be “they”. Moreover, the comparison was done between DPSCs and BM-MSCs. Authors wrote “the properties of DPSCs and MSCs”.
  2. Various review papers have been published regarding the therapeutic potential of Dental Pulp Stem Cells as listed below. What is the strong point of this review paper?
  3. Therapeutic potential of dental stem cells. J Tissue Eng. 2017 Jan-Dec; 8: 2041731417702531.
  4. Dental Tissue-Derived Mesenchymal Stem Cells: Applications in Tissue Engineering. Crit Rev Biomed Eng. 2018;46(5):429-468.
  5. Dental Pulp Stem Cell: A review of factors that influence the therapeutic potential of stem cell isolates. Biomaterials and Biomechanics in Bioengineering.
  6. Characteristics and Therapeutic Potential of Dental Pulp Stem Cells on Neurodegenerative Diseases. Front. Neurosci. 14:407. doi: 10.3389/fnins.2020.00407

  1. Line 67. Due to the immunomodulatory and anti-inflammatory properties and regenerative stem cells, research on their use in the treatment of COVID-19 is ongoing. Please provide a reference for this sentence.
  2. It is better if the paragraph from line 71 to 77 moved up after differentiation characterization.
  3. There is an age dependent decline of DPSCs in aged population. Maybe you need to discuss how researchers could improve their potentiality in aging whether it autologous or allogenic.
  4. IL-13, IL-6, TNF, HB-EGF, TGF-β1, TGF-β3, IGF-1, IGF-2, AGF. The complete name should be written in a complete form in first mention, then use it is abbreviation then. Authors should check throughout the manuscript. Line 154. mesenchymal stem cells however in line 44 it was abbreviated as MSCs!!!
  5. Line 113. Their reservoir is located around the perivascular niches in the pulp. Please provide ref. for this sentence.
  6. Line 170. This marker is a component of the growth factor-beta receptor (TGFbeta) complex…this sentence should be corrected in abbreviation and should be written in a complete form in first mention.
  7. in Line 198. PBS (phosphate-buffered saline) and again they wrote it in line 236 phosphate buffer saline (PBS).
  8. Line. 199 α-MEM mentioned in abbreviation for first time, then they mentioned full name after in line 223, and repeated at 243. This should be corrected throughout the manuscript.
  9. References 52, 54 should be corrected.
  10. Line 19; remove excess “and”.
  11. In 5.1. Cell Sources” and scaffold and growth factors and osteocyte differentiation and Xenotransplantation sections the references wrote by the authors names not citing numbers.
  12. English editing and proofreading through the entire manuscript with the assistance of an English-speaking colleague or use an English assistance company should be done.

Author Response

Dear Editor,

Thank you very much for the review of our Manuscript IDmolecules-1478891.

We sincerely appreciate all valuable comments and suggestions, which helped us to improve the quality of the article. Our responses to the Reviewer comment are described below in a point-to-point manner.

 Appropriated changes, suggested by the Reviewers, has been introduced to the manuscript such as change of title, details in some paragraphs etc. Moreover, several authors, along with her affiliation and contribution to the work, have been added.

1.The given title is misleading. Authors should focus on the in vivo therapeutic potential of DPSCs. However, they discussed a lot about the in vitro part?!!

Thank You for the suggestion. We would therefore like to propose a new title: Therapeutic Potential of Dental Pulp Stem Cells according to different transplant types.

2.Line 229. Authors mentioned that “The same method was used to isolate DPCs and DPSCs to 228 compare gene expression with MSCs and FBS [52]”

Which MSCs? I understand that FBS referred to Fetal Bovine Serum as you abbreviated elsewhere however the authors compare with fibroblasts??!!!

Suggestion accepted. This sentence has been changed.

3.Line 82. In fact, not only ADMSC and DPSC that are currently studies but other many sources, as you mentioned above, such as AFMSC, SFMSC, endometrial, umbilical…etc.

Thank you for the suggestion. This sentence has been expanded line 893.

4.Line 128, not only Gronthos who did the comparison but with other authors as well, “he” should be “they”. Moreover, the comparison was done between DPSCs and BM-MSCs. Authors wrote “the properties of DPSCs and MSCs”.

Your suggestion is very valuable. The sentence has been rewritten

5.Various review papers have been published regarding the therapeutic potential of Dental Pulp Stem Cells as listed below. What is the strong point of this review paper?

  1. Therapeutic potential of dental stem cells. J Tissue Eng. 2017 Jan-Dec; 8: 2041731417702531.
  2. Dental Tissue-Derived Mesenchymal Stem Cells: Applications in Tissue Engineering. Crit Rev Biomed Eng. 2018;46(5):429-468.
  3. Dental Pulp Stem Cell: A review of factors that influence the therapeutic potential of stem cell isolates. Biomaterials and Biomechanics in Bioengineering.
  4. Characteristics and Therapeutic Potential of Dental Pulp Stem Cells on Neurodegenerative Diseases. Front. Neurosci. 14:407. doi: 10.3389/fnins.2020.00407

Thank you for the suggestion. The work discusses the regenerative potential of various types of tissue transplants in a clear and detailed way. This approach is rarely seen in stem cell publications.

6.Line 67. Due to the immunomodulatory and anti-inflammatory properties and regenerative stem cells, research on their use in the treatment of COVID-19 is ongoing. Please provide a reference for this sentence.

Suggestion accepted.

Zayed, M., & Iohara, K. (2020). Immunomodulation and Regeneration Properties of Dental Pulp Stem Cells: A Potential Therapy to Treat Coronavirus Disease 2019. Cell Transplantation, 29, 1–9. https://doi.org/10.1177/0963689720952089

7.It is better if the paragraph from line 71 to 77 moved up after differentiation characterization.

Thank you for the suggestion but we decided to leave it in this place.

8.There is an age dependent decline of DPSCs in aged population. Maybe you need to discuss how researchers could improve their potentiality in aging whether it autologous or allogenic.

Thank you for the suggestion. A sentence and references has been added -line 129 . 

Changes in the dental pulp during ageing (decreased vascular supply, modification of the niches, obliteration of pulp chamber) affects function of DPSCs. These symptoms can affect the regenerative capacity of stem cells. Further studies of changes in dental pulp stem cells with age may play an important role in their use in the elderly.

Iezzi I, Pagella P, Mattioli-Belmonte M, Mitsiadis TA. The effects of ageing on dental pulp stem cells, the tooth longevity elixir. Eur Cell Mater. 2019 Feb 26;37:175-185. doi: 10.22203/eCM.v037a11. PMID: 30805914

9.IL-13, IL-6, TNF, HB-EGF, TGF-β1, TGF-β3, IGF-1, IGF-2, AGF. The complete name should be written in a complete form in first mention, then use it is abbreviation then. Authors should check throughout the manuscript. Line 154. mesenchymal stem cells however in line 44 it was abbreviated as MSCs!!!

Suggestion accepted. This sentence has been changed.

10.Line 113. Their reservoir is located around the perivascular niches in the pulp. Please provide ref. for this sentence.

Suggestion accepted.

11.Line 170. This marker is a component of the growth factor-beta receptor (TGFbeta) complex…this sentence should be corrected in abbreviation and should be written in a complete form in first mention.

Suggestion accepted.

12.in Line 198. PBS (phosphate-buffered saline) and again they wrote it in line 236 phosphate buffer saline (PBS).

Suggestion accepted.

13.Line. 199 α-MEM mentioned in abbreviation for first time, then they mentioned full name after in line 223, and repeated at 243. This should be corrected throughout the manuscript.

Suggestion accepted.

14.References 52, 54 should be corrected.

Suggestion accepted.

15.Line 19; remove excess “and”.

Suggestion accepted.

16. In 5.1. Cell Sources” and scaffold and growth factors and osteocyte differentiation and Xenotransplantation sections the references wrote by the authors names not citing numbers.

Your suggestion is very valuable. The references has been rewritten.

17.English editing and proofreading through the entire manuscript with the assistance of an English-speaking colleague or use an English assistance company should be done.

Thank you for this comment.  We would like to apologize for mistakes. We believe that English and grammar should not be a big issue that causes any difficulty in understanding the content.

We truly hope that the revised manuscript is now clarified enough for the reviewer to take the next step. We deeply appreciate your help to improve the readability of our paper.

Yours sincerely,

Anna Zawadzka-Knefel

Round 2

Reviewer 2 Report

  1. The authors mis cited reference no. (7) in the references list. It needs to be corrected.
  2. line 192, what means by DPDC markers? Do you mean DPSCs?
  3. line 171 and 181 mesenchymal stem cells should be written in abbreviated form.
  4. line 626-629, you mentioned eight reviewed studies, however you provided nine reviewed studies.
  5. 760&761 you don’t need to write the full name because you already mentioned it elsewhere.

Author Response

Dear Editor,

We would like to thank for your careful and thorough reading of this manuscript and for the thoughtful comments and constructive suggestions, which help to improve the quality of this manuscript. 

The manuscript has been revised as per the comments given by the reviewer, and our responses to all the comments are as follows:

1.The authors mis cited reference no. (7) in the references list. It needs to be corrected.

Your suggestion is very valuable. The correction has been made.

2.  line 192, what means by DPDC markers? Do you mean DPSCs?

Thank you for the correction. Instead of ,,DPCS markers" there should be ,,DPSCs markers''

3. line 171 and 181 mesenchymal stem cells should be written in abbreviated form.

Thank you so much for your minute observation. The correction has been made.

4. line 626-629, you mentioned eight reviewed studies, however you provided nine reviewed studies.

Thank you for the suggestion. The correction has been made.

5. 760&761 you don’t need to write the full name because you already mentioned it elsewhere.

Your suggestion is very valuable. The correction has been made.

We hope that these changes to the manuscript will facilitate the decision to publish this study in your journal. We have made a considerable effort to take into account the interesting suggestions proposed by the reviewers. In any case, we are open to consideration of any further comment on our answers.
Sincerely,
The authors